**DOI: 10.1038/ncomms13279**　　**OPEN**

# Excitonic linewidth and coherence lifetime in monolayer transition metal dichalcogenides

Malte Selig[1], Gunnar Berghäuser[1], Archana Raja[2,3], Philipp Nagler[4], Christian Schüller[4], Tony F. Heinz[3,5,6], Tobias Korn[4], Alexey Chernikov[4,6], Ermin Malic[7] & Andreas Knorr[1]

Atomically thin transition metal dichalcogenides are direct-gap semiconductors with strong light–matter and Coulomb interactions. The latter accounts for tightly bound excitons, which dominate their optical properties. Besides the optically accessible bright excitons, these systems exhibit a variety of dark excitonic states. They are not visible in the optical spectra, but can strongly influence the coherence lifetime and the linewidth of the emission from bright exciton states. Here, we investigate the microscopic origin of the excitonic coherence lifetime in two representative materials ($WS_2$ and $MoSe_2$) through a study combining microscopic theory with spectroscopic measurements. We show that the excitonic coherence lifetime is determined by phonon-induced intravalley scattering and intervalley scattering into dark excitonic states. In particular, in $WS_2$, we identify exciton relaxation processes involving phonon emission into lower-lying dark states that are operative at all temperatures.

[1] Institut für Theoretische Physik, Nichtlineare Optik und Quantenelektronik, Technische Universität Berlin, 10623 Berlin, Germany. [2] Department of Chemistry, Columbia University, New York, New York 10027, USA. [3] Department of Applied Physics, Stanford University, Stanford, California 94305, USA. [4] Institut für Experimentelle und Angewandte Physik, Universität Regensburg, 93040 Regensburg, Germany. [5] SLAC National Accelerator Laboratory, Menlo Park, California 94025, USA. [6] Departments of Physics and Electrical Engineering, Columbia University, New York, New York 10027, USA. [7] Chalmers University of Technology, Department of Physics, SE-412 96 Gothenburg, Sweden. Correspondence and requests for materials should be addressed to M.S. (email: malte.selig@campus.tu-berlin.de).

As truly two-dimensional (2D) materials exhibiting a weak dielectric screening, monolayer transition metal dichalcogenides (TMDs) show a remarkably strong Coulomb interaction giving rise to the formation of tightly bound excitons[1–5]. In addition to the optically accessible bright excitonic states located at the $K$ and $K'$ points at the corners of the hexagonal Brillouin zone[6–10], there is also a variety of optically forbidden states including $p$ excitons exhibiting a non-zero angular momentum, intravalley excitons with a non-zero centre-of-mass momentum beyond the light cone, spin-forbidden intravalley exciton triplets, as well as intervalley excitons where a hole is located at the $K$ point and the electron either at the $K'$ point or the $\Lambda$ point (half way between $K$ and $\Gamma$ point in the first Brillouin zone)[11–13], cf. Fig. 1. The corresponding transitions are denoted as $K - K'$ and $K - \Lambda$ in the following. In a recent time-resolved and temperature-dependent photoluminescence study, the existence of such dark intervalley excitons was experimentally demonstrated[14]. In particular, it was shown that in tungsten-based TMDs, the intervalley dark excitonic state lies energetically below the optically accessible exciton, resulting in a strong quenching of photoluminescence at low temperatures[14,15].

Since excitons dominate the optical response of TMDs, a microscopic understanding of their properties is crucial for technological applications in future optoelectronic and photonic devices. The presence of dark states has a strong impact on the coherence lifetime of optically accessible states, since they present a possible scattering channel that can be accessed via emission or absorption of phonons. The coherence lifetime of these excitonic states in 2D TMDs is directly reflected by the homogeneous linewidth of the corresponding excitonic resonances[16]. The homogeneous linewidth in monolayer WSe$_2$ has been recently measured via optical 2D Fourier transform spectroscopy, a

method which permits homogeneous broadening to be isolated from possible inhomogeneous broadening[17]. Moody et al. find a linear increase of the homogeneous linewidth for temperatures up to 50 K (refs 15,19,20). Further studies show both a linear and a super-linear increase of the homogeneous[18] and total linewidths at higher temperatures. The observed increase is typically ascribed to scattering with acoustic and optical phonons within the $K$ valley. Phonon-induced scattering into dark intervalley exciton states has not been considered to date. In addition, a consistent microscopic theory description of exciton–phonon scattering is not available yet.

Here, we present a joint theory–experiment study that aims to address the underlying microscopic processes determining the excitonic coherence lifetime and, thus, the intrinsic homogeneous linewidth in TMD monolayers. Our theoretical approach is based on the semiconductor Bloch equation for the microscopic polarization combined with the Wannier equation providing access to eigenvalues and eigenfunctions of the bright and dark excitons. The joint study reveals the existence of qualitatively different microscopic channels behind the excitonic coherence lifetime in tungsten- and molybdenum-based TMDs: in MoSe$_2$, the coherence lifetime is determined by radiative coupling at low temperatures and by phonon-induced intravalley scattering at room temperature. In contrast, the excitons in WS$_2$ can be efficiently scattered into the energetically lower-lying intervalley dark excitonic state, cf. Fig. 1. This process is driven by phonon emission and remains significant even at low temperatures. In this study, we focus on spin-conserving processes occurring on an ultrashort timescale (tens of femtoseconds). Intervalley spin–flip scattering occurs on a longer (picosecond[21,22]) timescale for supported samples. To describe the coherence lifetime of optically bright excitons, we develop a theoretical model that includes all relevant relaxation channels on a microscopic footing. In the experimental facet of the study, we tested the theory by characterizing WS$_2$ and MoSe$_2$ monolayers using linear reflectance and photoluminescence spectroscopy. From the experimental linewidths of the exciton transitions, we estimate the temperature-dependent homogeneous broadening of the resonances (additional details for the experimental procedure and data analysis are given in 'Methods' section). Our analysis is also consistent with recent reports on total exciton linewidths in MoSe$_2$ (ref. 15), as well as with the behaviour of WSe$_2$ and MoS$_2$ studied through coherent spectroscopy[17,18] and in optical-pump/midIR-probe experiments[23]. Moody et al.[17] also reported a linearly increasing homogeneous linewidth with the excitation density, which was attributed to exciton–exciton scattering. In this work we focus on the low-excitation regime, where these Coulomb induced processes play a minor role.

## Results

**Theoretical approach.** The first step of the theoretical analysis is the solution of the Wannier equation[3,16,24,25],

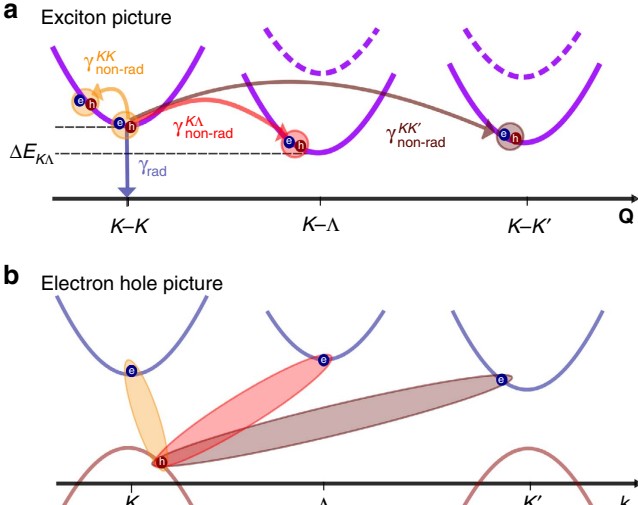

**Figure 1 | Relaxation channels determining the excitonic coherence lifetime.** (**a**) Schematic representation of the minima of the excitonic centre-of-mass motion (**Q**) dispersion $E(\mathbf{Q})$. A $K$-$K$ exciton can decay via radiative $\gamma_{\text{rad}}$ (blue arrow) or non-radiative dephasing $\gamma_{\text{non-rad}}$. The latter occurs through exciton–phonon scattering within the $K - K$ valley (orange) or to dark $K - \Lambda$ (red) or $K - K'$ (brown) excitonic states. For WS$_2$, the intervalley $K - \Lambda$ and $K - K'$ excitons lie energetically below the $K - K$ exciton ($\Delta E_{K\Lambda} < 0$) allowing efficient scattering via emission of phonons even at very low temperatures. The dashed dispersion curves refer to a situation typical in MoSe$_2$, where $\Delta E_{K\Lambda} > 0$. (**b**) A $K - K$ ($K - \Lambda$, $K - K'$) exciton is formed by a hole (red) at the $K$ point and an electron (blue) at the $K$ ($\Lambda, K'$) point.

$$\frac{\hbar^2 \mathbf{q}^2}{2m} \varphi_{\mathbf{q}}^{\mu} - \sum_{\mathbf{k}} V_{\mathbf{q},\mathbf{k}}^{\text{exc}} \varphi_{\mathbf{q}+\mathbf{k}}^{\mu} + E_{\text{gap}} \varphi_{\mathbf{q}}^{\mu} = E^{\mu} \varphi_{\mathbf{q}}^{\mu}, \qquad (1)$$

with the reduced mass of the exciton $m$, presenting an eigenvalue equation for excitons in TMDs. This equation includes the excitonic part of the Coulomb interaction $V_{\mathbf{q},\mathbf{k}}^{\text{exc}}$ which is treated within the Keldysh formalism for 2D systems[3,26], with an underlying substrate of dielectric constant of $\epsilon = 2.13$. We obtain excitonic eigenenergies $E^{\mu}$ and excitonic wavefunctions $\varphi_{\mathbf{q}}^{\mu}$ for optically allowed bright and optically forbidden dark excitons, enumerated by the quantum number $\mu$. The wavefunctions depend on the momentum $\mathbf{q} = \alpha \mathbf{k}_1 + \beta \mathbf{k}_2$ describing the relative motion of electrons ($\mathbf{k}_1$) and holes ($\mathbf{k}_2$) in real space, where $\alpha =$

$\frac{m_e}{m_h + m_e}$ and $\beta = \frac{m_h}{m_h + m_e}$ with effective masses for electrons and holes $m_e$, $m_h$.

The second step is to derive a Bloch equation for the microscopic polarization $p_{\mathbf{k}_1,\mathbf{k}_2}^{vc}(t)$ that determines the optical response of the material. The quantity reads in the excitonic basis[27] $P_{\mathbf{Q}}^{\mu} = \sum_{\mathbf{q}} \varphi_{\mathbf{q}}^{*\mu} \langle a_{\mathbf{q}+\beta\mathbf{Q}}^{\dagger v} a_{\mathbf{q}-\alpha\mathbf{Q}}^{c} \rangle$, with $a_{\mathbf{q}}^{(\dagger)\lambda}$ as annihilation (creation) operators for an electron in the state $(\mathbf{q},\lambda)$ with the band index $\lambda = (v,c)$ and $\mu$ being the quantum number of the exciton state. Here, we also have introduced the in-plane momentum $\mathbf{Q} = \mathbf{k}_1 - \mathbf{k}_2$ denoting the Fourier coordinate of the centre-of-mass motion in real space. The electronic dispersion is assumed to be parabolic, which is a good approximation in the vicinity of the $K$ point. This results in a quadratic dispersion for the excitons which constitutes the lowest excitonic contribution[11,12] addressed in a coherent optical transmission experiment with fixed polarization. We focus here on the energetically lowest lying intravalley and intervalley excitons numbered by $\mu$, that is, the bound electron and hole are either both located at the $K$ point (intravalley), or only the hole is at the $K$ point, while the electron is either in the $K'$ or in the $\Lambda$ point in the first Brillouin zone, cf. Fig. 1. Since an emitted or absorbed photon must match the centre-of-mass momentum $\mathbf{Q}$, only excitons with $\mathbf{Q} \approx 0$ are optically accessible. As a result, all intervalley excitons are dark.

Applying Heisenberg's equation of motion, $i\hbar\, \partial_t \mathcal{P}_{\mathbf{Q}}^{\mu} = [H, \mathcal{P}_{\mathbf{Q}}^{\mu}]$, we can determine the temporal evolution of the microscopic polarization $P_{\mathbf{Q}}^{\mu}$. The Hamilton operator $H$ includes: (i) an interaction-free part containing the dispersion of the electrons and phonons, (ii) the carrier–light interaction determining the optical selection rules, (iii) the carrier–carrier interaction that has already been considered in the Wannier equation and (iv) the carrier–phonon interaction coupling bright and dark excitons via emission and absorption of phonons. The carrier–light coupling is considered within the semi-classical approach in the $\mathbf{p} \cdot \mathbf{A}$ gauge[28]. The coupling is determined by the optical matrix element $M_{\mathbf{q}}^{\sigma-}$ projected to right-handed circular polarized light as required to excite $K-K$ excitons[3,29–31]. The carrier–phonon matrix element $g_q^{\lambda\alpha}$ is treated within an effective deformation potential approach for acoustic phonons and by approximating the Fröhlich interaction for optical phonons[32,33].

Evaluating the commutator in the Heisenberg equation of motion, we obtain within the second-order Born–Markov approximation[27] the Bloch equation for the microscopic polarizations $P_{\mathbf{Q}}^{\mu}(t)$:

$$i\hbar\, \partial_t P_{\mathbf{Q}}^{\mu}(t) = \left( \frac{\hbar^2 \mathbf{Q}^2}{2M} + E^{\mu} \right) P_{\mathbf{Q}}^{\mu} + i\hbar \sum_{\mathbf{q}} \varphi_{\mathbf{q}}^{*\mu} \Omega_{\mathbf{q}}^{cv} \delta_{\mathbf{Q},0} - i\frac{\pi}{\hbar} \sum_{\mathbf{q}',\alpha v} G_{\mathbf{Q},\mathbf{Q}+\mathbf{q}'}^{\mu v \alpha} P_{\mathbf{Q}}^{v}. \tag{2}$$

While the first term describes the oscillation of the excitonic polarization determined by the excitonic dispersion in $\mathbf{Q}$, the second term stands for the carrier–light interaction given by the Rabi frequency $\Omega_{\mathbf{q}}^{cv} = \frac{e_0}{m_0} M_{\mathbf{q}}^{\sigma-} A^{\sigma-}$ with the vector potential $A^{\sigma-}$ and the electron charge $e_0$ and mass $m_0$. The third contribution in equation (2) describes the exciton–phonon interaction that is given by the function

$$G_{\mathbf{Q},\mathbf{Q}+\mathbf{q}'}^{\mu v \alpha} = \hbar \sum_{\pm,\rho} g_{\mathbf{q}'}^{\mu\rho\alpha} g_{-\mathbf{q}'}^{\rho v \alpha} \left( \frac{1}{2} \pm \frac{1}{2} + n_{\mathbf{q}'}^{\alpha} \right) \times \delta\left( E^{\rho}(\mathbf{Q}+\mathbf{q}') - E^{v}(\mathbf{Q}) \pm \hbar\omega_{\pm\mathbf{q}'}^{\alpha} \right). \tag{3}$$

Here, $\hbar\omega_{\mathbf{q}'}^{\alpha}$ is the phonon energy and $n_{\mathbf{q}'}^{\alpha}$ the phonon occupation in the mode $\alpha$ corresponding to the Bose–Einstein distribution. The function contains scattering processes including phonon emission $(+)$ and absorption $(-)$. Overall, $G_{\mathbf{Q},\mathbf{Q}+\mathbf{q}'}^{\mu v \alpha}$ accounts for

scattering of an exciton from the state $\mu$ with momentum $\mathbf{Q}$ to the state $v$ with momentum $\mathbf{Q} + \mathbf{q}'$ under emission or absorption of a phonon in the mode $\alpha$ and momentum $\mathbf{q}'$. The exciton–phonon coupling

$$g_{\mathbf{q}'}^{\mu v \alpha} = \sum_{\mathbf{q}} \left( \varphi_{\mathbf{q}}^{*\mu} g_{\mathbf{q}'}^{c\alpha} \varphi_{q-\beta\mathbf{q}'}^{v} - \varphi_{\mathbf{q}}^{*\mu} g_{\mathbf{q}'}^{v\alpha} \varphi_{\mathbf{q}+\alpha\mathbf{q}'}^{v} \right) \tag{4}$$

depends on the electron–phonon matrix elements $g_{\mathbf{q}'}^{c\alpha}$ and $g_{\mathbf{q}'}^{v\alpha}$ for conduction and valence band and the overlap of the relevant exciton wavefunctions in momentum space. The corresponding phonon-induced homogeneous dephasing of the excitonic polarization reads[27]

$$\gamma_{\mathbf{Q}}^{\mu\alpha} = \frac{\pi}{\hbar^2} \sum_{\mathbf{q}'v} G_{\mathbf{Q},\mathbf{Q}+\mathbf{q}'}^{\mu v \alpha} \tag{5}$$

giving rise to a non-radiative coherence lifetime for excitons with momentum $\mathbf{Q}$ in the state $\mu$. Here, we take into account acoustic (LA, TA) and optical phonons (LO, TO)[33], explicitly considering intravalley scattering between bright ($\mathbf{Q} = 0$) and dark $K-K$ excitonic states ($\mathbf{Q} \neq 0$) as well as intervalley scattering involving dark $K-\Lambda$ and $K-K'$ excitons, cf. Fig. 1. To evaluate the exciton–phonon scattering rates, we calculate the rate self-consistently, which corresponds to a self-consistent Born approximation[34,35].

Besides the exciton–phonon scattering, the coherence lifetime of excitons is also influenced by radiative coupling, that is, spontaneous emission of light through recombination of electrons and holes. The radiative coupling is obtained by self-consistently solving the Bloch equation for the excitonic polarization and the Maxwell equations in a 2D geometry for the vector potential $A^{\sigma-}$ yielding[36]

$$\gamma_{\mathrm{rad}} = \frac{\hbar^2 c \mu_0}{\omega n} \left| \sum_{\mathbf{q}} M_{\mathbf{q}}^{\sigma-*} \varphi_{\mathbf{q}}^{\mu} \right|^2. \tag{6}$$

Here, $c/n$ is the light velocity in the substrate material and $\mu_0$ the vacuum permeability.

**Excitonic absorption spectra.** Having solved the Wannier equation for $K-K$, $K-\Lambda$ and $K-K'$ excitons, equation (1), and the Bloch equation, equation (2), we have access to the optical response of TMDs and can evaluate the radiative and non-radiative homogeneous linewidth of excitonic resonances to obtain the coherence lifetime of optically allowed excitons. Figure 2 illustrates the absorption spectrum of 2D sheets[36] of two representative monolayer TMDs including tungsten diselenide (WS$_2$) and molybdenum diselenide (MoSe$_2$). We predict the homogeneous linewidth of the energetically lowest lying A exciton to be in the range of a few meV corresponding to an excitonic coherence lifetime of a few hundreds of femtoseconds. Depending on the temperature, either the radiative or the non-radiative contribution is the dominant mechanism. We observe a larger radiative broadening in WS$_2$ (7 versus 4 meV for MoSe$_2$), while the overall broadening at room temperature is larger for MoSe$_2$, namely, 40 versus 24 meV in WS$_2$. Furthermore, Fig. 2 shows the temporal evolution of the excitonic polarization $P_0^{1s}(t)$ after optical excitation with a 10 fs pulse. We find that the polarization decays radiatively with a time constant in the range of hundreds of fs. Including exciton–phonon coupling the time constant decreases sharply to tens of fs at room temperature. Evaluating equations (5) and (6), we can determine the microscopic origin of the excitonic linewidth. Figure 3 shows the temperature dependence of the different contributions to the total linewidth. Further, we show the corresponding exciton coherence lifetime which is connected by $L\tau = \hbar$, with $L$ being the

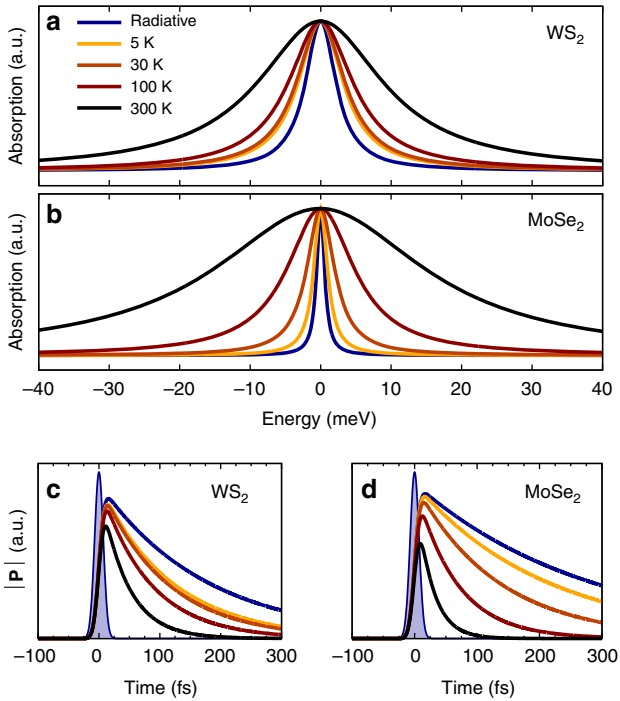

**Figure 2 | Calculated homogeneous broadening and coherence lifetime.** Predicted absorption spectrum of (**a**) WS$_2$ and (**b**) MoSe$_2$ for the energetically lowest resonance of the A exciton. All spectra were normalized to have the same peak value. While the blue line only includes the radiative linewidth, the other lines also contain non-radiative contributions due to exciton–phonon scattering at different temperatures. The magnitude of the microscopic polarization for (**c**) WS$_2$ and (**d**) MoSe$_2$ following excitation by a 10 fs pulse at $t = 0$ (filled curve).

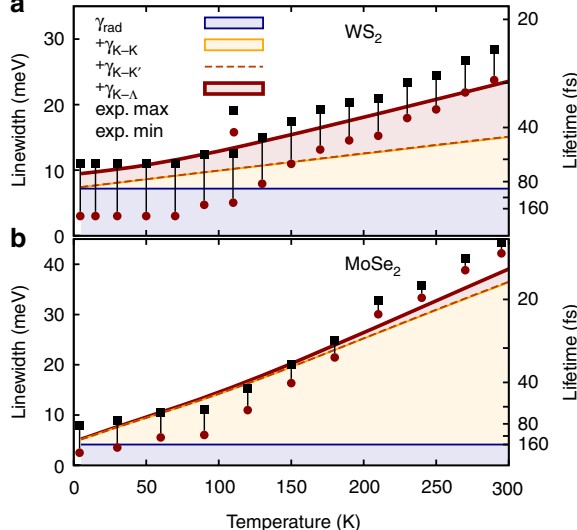

**Figure 3 | Excitonic linewidth and lifetime.** Temperature dependence of the linewidth and lifetime of the A exciton in (**a**) WS$_2$ and (**b**) MoSe$_2$. The red and black points correspond to the minimum and maximum limits of the homogeneous linewidths extracted from experiment, respectively (see Methods section). The thick red line shows the total predicted linewidth consisting of the single contributions from the radiative decay $\gamma_{rad}$ (blue) and non-radiative decay from intravalley exciton–phonon coupling $\gamma_{non-rad}^{K\Lambda}$ (orange) and intervalley coupling $\gamma_{non-rad}^{KK}$ and $\gamma_{non-rad}^{KK'}$ (dashed orange). Note that the latter contribution is very small.

full linewidth. We find an excellent agreement between theory and experiment with respect to qualitative trends and reasonable quantitative agreement of the linewidths.

**Temperature-dependent excitonic linewidth.** In both investigated TMD materials, we observe a temperature-independent offset originating from radiative recombination. Our results for the radiative dephasing are in good agreement with recent calculations[17,37]. In contrast, the non-radiative coupling via scattering with phonons introduces a strong temperature dependence. Furthermore, we find a remarkably different behaviour for MoSe$_2$ and WS$_2$: while for MoSe$_2$ intravalley exciton–phonon scattering is the crucial mechanism, the exciton linewidth in WS$_2$ is dominated by intervalley scattering $\gamma_{non-rad}^{K\Lambda}$, coupling the optically allowed $K-K$ exciton with the dark $K-\Lambda$ states. The reason for this difference lies in the relative energies of these excitons. In WS$_2$, the $K-\Lambda$ exciton is located $\sim 70$ meV below the $K-K$ exciton. Hence, exciton relaxation through phonon emission is very efficient even at 0 K, resulting in a non-radiative contribution to the homogeneous linewidth at low temperatures, cf. Fig. 3a. Zhao et al.[38] report a change of the band structure with temperature due to an extension of the lattice, which is not included here. According to ref. 38 one would observe a larger separation between $K-K$ and $K-\Lambda$ excitons at lower temperatures, which would increase the non-radiative contribution to the homogeneous linewidth. However, this effect is expected to be small, since the radiative linewidth dominates at low temperatures. The coupling to $K-K'$ excitons does not significantly contribute to the exciton linewidth due to the weak electron–phonon coupling element[33].

The situation is entirely different in MoSe$_2$, where the dark $K-\Lambda$ exciton lies $\sim 100$ meV above the bright $K-K$ exciton. Thus, only the less efficient absorption of phonons can take place. Since the energy of large-momentum acoustic phonons, required for scattering $K-K$ excitons into $K-\Lambda$ states, is only 15 meV in this material[33], the only contribution of intervalley scattering in MoSe$_2$ stems from the absorption of optical $\Gamma$ phonons and becomes relevant for temperatures higher than 150 K, cf. Fig. 3b. Our calculations show that the intravalley scattering with acoustic $\Gamma$ phonons is crucial for the excitonic coherence lifetime in MoSe$_2$ resulting in a nearly linear increase of linewidth with temperature, as reported for monolayer MoTe$_2$ (ref. 20) and MoS$_2$ (ref. 18). Since acoustic $\Gamma$ phonons have low energies, the Bose–Einstein distribution appearing in equation (3) can be linearized resulting in a dephasing rate $\gamma_{non-rad,ac}^{KK'} = 2\pi^2 |g_{q_0}|^2 \frac{kT}{2\hbar M c_{ac}^2}$ exhibiting a linear dependence on the temperature $T$. The slope is given by the exciton mass $M$, the velocity of the acoustic phonons $c_{ac}$ and the exciton–phonon coupling element $g_{q0}$ at the position $q_0$, where the delta distribution is fulfilled, cf. equation (3).

The additional super-linear increase of the linewidth with temperature observed for both MoSe$_2$ and WS$_2$ is ascribed to the scattering of excitons with optical $\Gamma$ or zone-edge $\Lambda$ phonons. The overall microscopically calculated temperature dependence of the linewidth can be phenomenologically approximated by $\gamma = \gamma_0 + c_1 T + \frac{c_2}{e^{\frac{\Omega}{kT}} - 1}$. For WS$_2$, we obtain a temperature-independent offset of $\gamma_0 = 9.2$ meV (consisting of 7 meV due to radiative decay and 2.1 meV due to acoustic $\Lambda$ phonon emission), a slope $c_1 = 28\,\mu eV\,K^{-1}$ describing the linear increase due to acoustic $\Gamma$ phonons, a rate $c_2 = 6.5$ meV and an averaged energy $\Omega = 20$ meV of relevant acoustic $\Lambda$ (that is, zone-edge) phonons defining the superlinear increase. We find that optical $\Gamma$ phonons do not give a strong contribution to the superlinear increase, since the coupling element in equation (4) has nearly

vanished at the required large momenta due to the sharper exciton wavefunctions in $WS_2$. The corresponding parameters for $MoSe_2$ read $\gamma_0 = 4.3$ meV, $c_1 = 91$ µeV K$^{-1}$, $c_2 = 15.6$ meV (7.2 meV due to intravalley optical phonon scattering and 8.4 meV due to $K - \Lambda$ coupling) and $\Omega = 30$ meV. Overall, our findings correspond well to the observations in refs 15 and 18, emphasizing the importance of thermally activated scattering processes with high-momentum phonon-modes beyond the linear acoustic regime. Also, since the relative position of the dark and bright exciton states plays a crucial role, TMDs with the same transition metal are generally expected to show a similar behaviour (see Supplementary Fig. 1 and Supplementary Note 1 for measured and computed data for $WSe_2$). Finally, since the presence of the substrate has an impact on the exciton wavefunctions due to the dielectric screening, the exciton–phonon coupling elements $g_{q'}^{\mu\nu\alpha}$ (equation (4)) depend on the choice of the substrate as well. Overall, both the radiative rate and exciton–phonon scattering are expected to decrease with increasing dielectric constant of the substrate.

## Discussion

We have presented a joint theoretical and experimental study revealing the microscopic origin of the excitonic lifetime in atomically thin 2D materials. We find both in theory and experiment a qualitatively different origin of the coherence lifetime limiting processes in tungsten- and molybdenum-based TMDs. While in $MoSe_2$, the coherence lifetime of an optically bright exciton is determined by the radiative rate and intravalley scattering with acoustic phonons, in $WS_2$ scattering into dark excitonic $K - \Lambda$ states is crucial. The gained insights shed light into excitonic properties that are crucial for exploiting the technological potential of these atomically thin nanomaterials. In particular, it will allow us to access exciton dynamics on a microscopic level across a large variety of relevant experimental scenarios, including exciton formation, thermalization and relaxation among many others. The presented theoretical approach can be furthermore generalized to quantitatively describe exciton behaviour for the whole family of semiconducting 2D materials beyond the representative systems studied here and provides a theoretical basis to explore fundamental many-body physics of these systems, crucial for future applications and allowing for consistent theoretical predictions of the functionality for novel devices on inter-atomic scales.

## Methods

In our theoretical treatment, all matrix elements were calculated within the tight-binding approximation[3]. The required parameters for the bandgap energy, the spin–orbit splitting, the effective masses, phonon energies and strength of electron–phonon coupling were taken from density functional theory (DFT) calculations[33,39]. Furthermore, we assume a constant energy for optical phonons and acoustic phonons with large wave numbers, that is, $\hbar\omega_q^\alpha =$ const (Einstein model) and a linear dispersion for acoustic phonons in the long wavelength limit, that is, $\hbar\omega_q^\alpha = \hbar c_{ac}^\alpha q$ (Debye model) with the mode-dependent velocity $c_{ac}^\alpha$. The oscillator strength of the optical matrix element is adjusted to the experimentally measured absorption maximum of the A exciton[40].

In the experimental study, $WS_2$ and $MoSe_2$ monolayer samples were obtained by mechanical exfoliation of the respective bulk crystals on fused silica and $SiO_2/Si$ substrates. The samples were held in an optical cryostat and studied in the temperature range between 4 and 300 K. Broadband radiation from a tungsten-halogen lamp and the 532 nm radiation of a continuous-wave laser source were used, respectively, for reflectance and photoluminescence measurements. The light was focused on the samples by a microscope objective, resulting in spot sizes with a diameter of a few micrometres. Both the reflected radiation and the photoluminescence signal were dispersed in a spectrometer and subsequently detected by a CCD. In the reflectance measurements, the data were analysed by taking into account the modification of the optical response by the underlying substrate. As the temperature was increased from 4 to 300 K, the exciton transitions broadened and shifted to lower energies; the area under the resonance remained unchanged (within our 20% accuracy). Equivalent results for the exciton peak width and position were obtained in photoluminescence measurements. In our

experiment, we measured temperature-dependent total linewidth $\Gamma_{tot}(T)$ of the ground state exciton resonance of $WS_2$ and $MoSe_2$. The measured linewidth is considered as a convolution of the intrinsic homogeneous linewidth $\Gamma_h(T)$ with potential inhomogeneous broadening $\Gamma_i$ from residual defects and disorder. The inhomogeneous broadening was assumed not to depend on temperature and was estimated from the low-temperature homogeneous linewidth $\Gamma_h(T=0)$ in the studied materials. The lowest possible $\Gamma_h(T=0)$ is determined by the radiative lifetime of the exciton. We obtain the radiative lifetime using its relation to the oscillator strength[21,41], the latter being calculated from the measured imaginary part of the dielectric function $\varepsilon$ of the sample. The maximum value of $\Gamma_h(T=0)$ was determined by the smallest measured total linewidths in $WS_2$ and $MoSe_2$ of 11 and 8 meV, respectively. These two limits thus fixed the maximal and minimal inhomogeneous broadening $\Gamma_i$ in the measured samples and allowed us to extract temperature-dependent homogeneous linewidths from the experimental data. The homogeneous contribution was considered to be a Lorentzian and the inhomogeneous contribution to be of Gaussian shape. We combined the linewidths using an analytical approximation for the convolution[42].

**Data availability.** The data that support the findings of this study are available from the corresponding author upon request.

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

## Acknowledgements

The authors thank Maja Feierabend and Mikhail Glazov for fruitful discussions. We acknowledge financial support from the Deutsche Forschungsgemeinschaft (DFG) through SFB 951 (G.B. and A.K.) SFB 787 (M.S.), SFB 689 (T.K. and C.S.), GK 1570 (P.N.), Emmy Noether Program (A.C.) and the EU Graphene Flagship (CNECT-ICT-604391) (E.M.,G.B.). At Columbia University support was provided by the Center for Precision Assembly of Superstratic and Superatomic Solids, an NSF MRSEC through Grant No. DMR-1420634 (A.R.) and by the W. M. Keck Foundation (A.C.). At SLAC/Stanford, this work was supported by the U.S. Department of Energy, Office of Science, Office of Basic Energy Sciences through the AMOS program within the Chemical Sciences, Geosciences, and Biosciences Division and by the Gordon and Betty Moore Foundation s EPiQS Initiative through Grant No. GBMF4545 (T.F.H.).

## Author contributions

M.S., G.B., E.M. and A.K. developed the microscopic theoretical model. A.R., P.N., T.K. and A.C. performed the experiments. All authors contributed to the interpretation of the results and writing of the manuscript.

## Additional information

**Competing financial interests:** The authors declare no competing financial interests.

