## [Peer Review File · Nature Communications]

Reviewers' comments:

Reviewer #1 (Remarks to the Author):

The authors present primarily theoretical work that seeks to provide a complete microscopic picture of the dephasing mechanism and as a result, the homogeneous line width in TMDs. This is the most complete and well thought out theoretical modeling of the dephasing processes in TMDs that has been published recently, which best explains recently published experimental results.

Applying the Heisenberg's equation of motion, they determine the temporal evolution of the microscopic polarization. In the Hamiltonian they include the interaction-free part which contains the dispersion of electrons and phonons, the carrier light interaction determining the optical selection rules, the carrier-carrier interaction included in the Wannier equation, and the carrier phonon interaction coupling bright and dark excitons via absorption and emission of phonons. They include intra and intervalley exciton-phonon scattering between bright and dark excitons. Therefore, the authors present a very robust model which includes all the possible interaction mechanisms.

There have been recently two direct measurements of the coherence time of excitons in TMDs, namely Ref. 15 and 17 in the manuscript. The present theoretical work solves a very interesting discrepancy between two published experimental papers. Ref 15 presents measurements of the excitonic dephasing or homogeneous linewidth on a CVD grown WSe₂ sample and concludes that the coherence is limited by the population time. This is also supported by Ref. 20. Whereas Ref. 17 reports on measurements performed on exfoliated samples of MoS₂, MoSe₂, and WSe₂ and conclude that the dephasing time is limited by exciton-phonon interactions. The present work determines that both conclusions are correct.

The authors conclude based on their theoretical modeling that in MoSe₂ the dephasing mechanism at low temperature is indeed limited by the radiative population time. However, in WS₂ and probably MoS₂ it is indeed limited by exciton-phonon scattering to lower dark excitonic states. They also conclude that this scattering is very efficient even at zero Kelvin. It would be indeed very interesting to have calculations for WSe₂ in order to have certainty whether it follows the MoSe₂ or the WS₂ mechanism. However, this should not affect or delay the publication of the manuscript.

Finally, it would further strengthen the manuscript if the authors discussed a bit more in detail the experimental results of Ref 17, since these results support their conclusions. In Ref 17 three different materials are compared directly and they are exfoliated samples, which tend to exhibit higher material quality. The dephasing trend between the materials follows the theory quite well, with MoS₂ showing the stronger dephasing and strong temperature dependence at low temperatures, indicating phonon-assisted dephasing processes. The authors mention that MoS₂ should have a similar excitonic structure as WS₂, which would justify this behavior.

The MoSe₂ sample on the other hand exhibits clearly the longest dephasing time of all of these materials studied there, which could likely be limited by the population time. WSe₂ lies in between and could follow either of the two schemes depending on the energy position of the dark exciton state.

Small error: In reference 23 the word 'graphane' should be 'graphene'.

In conclusion, the authors present a robust theoretical model that explains well recent experimental results. Therefore, I enthusiastically recommend publication of the manuscript.

Reviewer #2 (Remarks to the Author):

In this manuscript the authors report their study on homogeneous excitonic linewidth in monolayer of WS₂ and MoSe₂. The work covers joint theoretical and experimental analysis of relaxation lifetimes of ground excitons in two different systems. The results provide yet another proof of difference in exciton decay dynamics in Mo- and W-based atomically thin dichalcogenides. Although similar discussions have been drawn in previous works, (e.g. G. Moody et al in Nat. Comm. 6:8315, 2015; and Zhang et al. in PRL 115:257403, 2015) current manuscript reveal a detailed picture of underlying mechanisms of exciton relaxation pathways with great support from theoretical approach.

I believe this work will be of interest to the readers of Nature Communications. The manuscript is well-written and well-organized with the clear logical flow. In the following however I would like to address few topics to which the authors may comment in the manuscript.

(i) Could the authors comment on exciton-exciton and exciton-free carrier induced dephasing and its impact into the whole picture of excitonic lifetimes?

(ii) Electronic structure of dichalcogenides is known to change with temperature. These changes might lead to an alternation of relative energy gap between K and Λ valley minima in conductance band, for example (see e.g. Zhao et al in Nano Lett. 13: 5627, 2013). Could authors comment on how these changes may affect the linewidth of the considered systems?

Reviewer #3 (Remarks to the Author):

This manuscript discusses the excitonic linewidth of monolayer transition metal dichalcogenides. Monolayer WS₂ and MoSe₂ are used as model materials for two different types of semiconducting TMDs: TMDs with dark exciton energies reside below (above) the bright exciton energies. The major conclusion is that, the coherence lifetime in monolayer MoSe₂ is determined by the intravalley scatterings with acoustic phonons, but in monolayer WS₂, the phonon scatterings into the dark excitonic states are efficient even at 0 Kelvin because these dark excitonic states have lower energies compared to the bright ones.

I find this manuscript is well written, and the presentation of the joint theoretical-experimental study is clear and convincing. Hence I recommend publishing this manuscript with minor revisions:

[1] Could the authors provide additional information of the Wannier excitons used in the theoretical model? What are the exciton radii used in the calculation, and are values consistent with the previous publications?

[2] In Figure 3, how reliable are the linewidths calculated from Kramers-Kronig based on the reflectance measurements given that the samples were exfoliated onto SiO₂/Si substrates? Are the measured integrated exciton oscillator strengths conserved quantities (temperature invariant)?

[3] How was the dielectric environment (Air/TMD/SiO₂) included in the model? By effective dielectric constant? Could the authors discuss the effect of the substrate on the excitonic linewidth?

Referee 1:

First of all, we are glad about the very positive evaluation and that the referee strongly recommends our work for Nature Communication. We have followed his/her advice and have addressed all raised points:

1. Comment:

It would be indeed very interesting to have calculations for WSe2 in order to have certainty whether it follows the MoSe2 or the WS2 mechanism.

Reply:

We thank the referee for this important comment and have carried out the corresponding calculations. The excitonic linewidths strongly depends on the relative energetic position between the bright and the dark excitonic states. Therefore, we find a clear asymmetry between the molybdenum- and the tungsten-based TMD family: For the first, the dark exciton states at the Λ valley are located energetically above the bright states, which leads to inefficient phonon-induced scattering into these states, cf. Fig. 1 in the main manuscript. In contrast, in tungsten-based TMDs the dark Λ valley states lie energetically below the optically bright excitons resulting in efficient exciton-phonon scattering. As suggested by the referee, we have also performed calculations for the homogeneous linewidth in WSe2, cf. Fig R1, where we find a clear impact of intervalley scattering into the dark Λ valley excitons – in line with the behavior observed in WS2.

Changes:

We have included a sentence on page 6 in the main manuscript addressing the behavior in different TMD families. “Since the relative position of the dark and bright exciton states plays the crucial role, TMDs with the same transition metal show a similar behavior.”

Fig. R1: Computed homogeneous linewidth in WSe2 corresponding to the behavior in WS2 described in the main text

2. Comment:

Finally, it would further strengthen the manuscript if the authors discussed a bit more in detail the experimental results of Ref 17, since these results support their conclusions. In Ref 17 three different materials are compared directly and they are exfoliated samples, which tend to exhibit higher material quality. The dephasing trend between the materials follows the theory quite well, with MoS2 showing the stronger dephasing and strong temperature dependence at low temperatures, indicating phonon-assisted dephasing processes. The authors mention that MoS2 should have a similar excitonic structure as WS2, which would justify this behavior.

Reply:

We thank the referee for the suggestion. While we cannot address the data for bulk materials, the observations for the monolayer broadening in Ref. [17] are indeed highly relevant for this work. Both the linear increase of the linewidth at low temperatures and the super-linear contribution reported in Ref. [17] are reproduced by the theory. In addition, the extracted phonon energies from the temperature-dependent broadening in Ref.[17] are also similar to our results.

Changes:

(Note, that since we had added a additional citation Ref 14, Arora et al., all citations shifted by 1 in the new version.)

We have thus included a more specific citation of the Ref. [18(formerly 17)] in the discussion part of the manuscript. In particular, we emphasized the importance of high-momentum phonon-modes at finite energies of several 10's of meV, beyond the linear acoustic range of the phonon dispersion. "Overall, our findings correspond well to the observations in Refs.[13, 18], emphasizing the importance of thermally-activated scattering processes with high-momentum phonon-modes beyond the linear acoustic regime. Also, since the relative position of the dark and bright exciton states plays a crucial role, TMDs with the same transition metal are generally expected to show a similar behavior."

3. Comment:

Small error: In reference 23 the word 'graphane' should be 'graphene'.

Reply:

Ref 23 treats the screening in two dimensional sheets, in this case the authors study indeed graphane, a derivative of graphene. Since graphane and TMDs are both thin films with finite bandgaps, the way of screening the Coulomb potential in both systems is comparable.

Referee 2:

We thank the referee for the positive evaluation of our work and the statement that it is of interest for the readership of Nature Communications.

1. Comment:

Could the authors comment on exciton-exciton and exciton-free carrier induced dephasing and its impact into the whole picture of excitonic lifetimes?

Reply:

In our manuscript, we focus on the weak excitation regime in slightly doped samples, where the linewidth is dominated by exciton-phonon scattering. However, as the referee states, exciton-exciton and exciton-electron scattering present another broadening mechanism at more elevated excitations or in gated samples. To see such effects, a dependence of the linewidth on the excitation or on gate voltage must be observed. For instance, in Ref 16(formerly 15), a first experimental study on the dependence of the homogeneous linewidth on the pump excitation fluence has been presented. The authors found a linearly increasing linewidth, which they attributed to exciton-exciton Coulomb scattering. The investigation of these channels that play an important role in the strong excitation regime will be in the focus of a future work.

Changes:

We have added a paragraph on page 2 summarizing the main results from Ref 16(formerly 15) on the possible impact of exciton-exciton scattering on the excitonic linewidth. "Reference [16] also

reported on a linearly increasing homogeneous linewidth with the excitation density, which was attributed to exciton-exciton scattering. In this work we focus on the low-excitation regime, where these Coulomb-induced processes play a minor role.”

2. Comment:

Electronic structure of dichalcogenides is known to change with temperature. These changes might lead to an alternation of relative energy gap between K and Λ valley minima in conductance band, for example (see e.g. Zhao et al in Nano Lett. 13: 5627, 2013). Could authors comment on how these changes may affect the linewidth of the considered systems?

Reply:

This is a very interesting point. Zhao et al. find that the relative energetic position of the conduction band minima change under increase of temperature due to the extension of the lattice. They find that the relevant separation of the K and the Λ valley increases linearly by 30 meV from 0 to 300 K. This of course may affect the efficiency of exciton-phonon scattering into the dark K- Λ states. In particular, we expect the energy separation between the bright K-K and the dark K- Λ excitons to decrease with temperature. In our calculations, we have included room-temperature DFT parameters for the electronic bandstructure from Ref 36 (formerly 35). As a result, the separation between the K- Λ exciton and the K-K exciton is expected to be larger at smaller temperatures, which would result in a more efficient exciton-phonon scattering from the K-K state into the energetically lower lying K- Λ state. Note however that this effect is small compared to the radiative contribution which is the dominant relaxation channels at low temperatures. In summary, we expect that including temperature-dependent changes in the electronic bandstructure, the homogeneous linewidth should only slightly increase at lower temperatures.

Changes: We added a paragraph to that point and reference the work of Zhao et al. on page 5. “Reference 39 reports a change of the band structure with temperature due to an extension of the lattice, which is not included here. According to Ref. 39 one would observe a larger separation between K-K and K- Λ exciton at lower temperatures, which would increase the non-radiative contribution to the homogeneous linewidth. However, this effect is expected to be small, since the radiative linewidth dominates at low temperatures.”

Referee 3:

We thank the referee for the careful evaluation of our manuscript. His/her comments have in particular improved the discussion on the substrate dependence of the homogeneous linewidth.

1. Comment:

Could the authors provide additional information of the Wannier excitons used in the theoretical model? What are the exciton radii used in the calculation, and are values consistent with the previous publications?

Reply:

The referee asks the relevant question, whether our approach is consistent with earlier works: Exciton wavefunctions as well as the exciton binding energies were obtained as a solution of the Wannier equation, being consistent with recent publications, cf. Ref 3. There, the consistence with experimental data has also been shown.

Fourier transforming our exciton wavefunction into the real space, we can determine a FWHM of 1.2 nm, which is consistent with Ref. 15, Mouri et al. Phys. Rev. B **90**, 155449(2014), and Cheiwchanchamngij, Phys. Rev. **B85**, 205302(2012).

2. Comment:

In Figure 3, how reliable are the linewidths calculated from Kramers-Kronig based on the reflectance measurements given that the samples were exfoliated onto SiO₂/Si substrates? Are the

measured integrated exciton oscillator strengths conserved quantities (temperature invariant)?

Reply:

To extract the linewidths from reflectance measurements we parameterize the dielectric function with Lorentzians and the optical response is then calculated using transfer-matrix method. The peak parameters are then adjusted to match the experiment. Thus, the changes in the optical response due to the substrate are properly taken into account. In addition, since we are addressing strong ground state exciton resonances at the fundamental band edge, the contributions from higher energy transitions (B-exciton, C-band, etc.) are rather small.

The oscillator strengths are indeed almost constant as the temperature is increased. In our analysis the samples tend to exhibit a slight decrease within 10-20% towards room-temperature, yet the mechanism is still under investigation (possible transfer of oscillator strength to phonon-sidebands, etc.).

Changes: We added a short paragraph to the methods section to address these points: “In reflectance, the data was analyzed by taking into account the modification of the optical response by the underlying substrate. In addition, as the temperature increased from 4 K to 300 K, the changes of the exciton transition in reflectance signal were largely due to the peak broadening and the shift of the peak position to lower energies; the area under the resonance remained almost invariant (within roughly 20% accuracy).

3. Comment:

How was the dielectric environment(Air/TMD/SiO₂) included in the model? By effective dielectric constant? Could the authors discuss the effect of the substrate on the excitonic linewidth?

Reply:

This is an important point. The Coulomb potential in the monolayer with dielectric environment was treated as a Keldysh potential, where we can take into account the interface between air and a fused silica substrate sandwiching the monolayer (corresponding to the investigated sample in our experiments). To estimate the effect of the substrate on the linewidth we have to look on several aspects:

(i) To determine exciton binding energies and exciton wavefunctions, we evaluate the Wannier equation using the above mentioned Keldysh potential. Thus both binding energies and wavefunctions are affected by the dielectric environment. Higher dielectric constants of the substrate yield lower exciton binding energies and more peaked exciton wavefunctions in momentum space. This has an impact on the radiative decay and the exciton-phonon scattering rates.

(ii) The exciton wavefunctions enter in the exciton-phonon coupling element. Since with increasing dielectric constant the width of the wavefunction in momentum space decreases, the coupling elements as well become sharper in the momentum domain. This leads to a suppression of exciton phonon scattering for higher dielectric constants.

(iii) The exciton binding energies, determining the relative position of the exciton states depend on the substrate. We observe a stronger redshift of the K-K excitons compared to the K- Λ excitons, which yields a less efficient intervalley scattering at higher dielectric constants.

(iv) The radiative decay rate depends on the mean refractive index of the materials located below and above the TMD layer (here air and fused silica). We find that the radiative coupling decreases with the substrate dielectric constant, following a power law behavior.

To sum up, both the radiative and the non-radiative contribution to the homogeneous linewidth decreases with an increasing dielectric constant of the substrate, which will be one of the directions

for further studies.

Changes: We added a short paragraph on that in the manuscript.

“Since, the presence of the substrate has an impact on the exciton wavefunctions due to the dielectric screening, the exciton-phonon coupling elements $g^{i\nu a}$ (Eq. 4) depend on the choice of the substrate as well. Overall, both the radiative broadening and the exciton-phonon scattering are expected to decrease with increasing dielectric constant of the substrate.”

REVIEWERS' COMMENTS:

Reviewer #1 (Remarks to the Author):

The authors have addressed all of the criticism raised by the referees satisfactorily. My recommendations were rather optional and should not have further delayed the publication of the manuscript. However, I feel that the authors have responded well to the rather minor but very valid points raised by the other two referees. Therefore, I feel that at this point the manuscript should be accepted for publication.

Reviewer #2 (Remarks to the Author):

The recent revision of the manuscript made the work to be more solid and comprehensive. Detailed analysis of the suggested topics provided by the authors complete the study. I am certain the work will be of a great interest within the readers of the community and so it deserve to be published in Nature Communications.

I am suggesting to include to the main text (or supplementary info file) the Fig.R1 (results for WSe₂ monolayer) presented in regards to the question of the first referee. This figure compares computed linewidth with experimental results that, together with data presented in the main text, evidence clear difference in excitonic behavior of W- and Mo-based monolayers. No need for another galley proof. I approve the publication.

Reviewer #3 (Remarks to the Author):

The revised manuscript and the response letter have addressed all the questions in the last review. The revised manuscript could be published as is.